# Integrating quantitative and qualitative data and findings when undertaking randomised controlled trials

David A Richards [1], Patricia Bazeley,[2] Gunilla Borglin,[3,4] Peter Craig,[5] Richard Emsley,[6] Julia Frost [1], Jacqueline Hill,[1] Jeremy Horwood,[7] Hayley Anne Hutchings [8], Clare Jinks,[9] Alan Montgomery,[10] Graham Moore,[11] Vicki L Plano Clark [12], Sarah Tonkin-Crine,[13] Julia Wade,[7] Fiona C Warren,[1] Sally Wyke,[14] Bridget Young,[15] Alicia O'Cathain [16]

**Correspondence to**
Professor David A Richards;
D.A.Richards@exeter.ac.uk

## ABSTRACT

It is common to undertake qualitative research alongside randomised controlled trials (RCTs) when evaluating complex interventions. Researchers tend to analyse these datasets one by one and then consider their findings separately within the discussion section of the final report, rarely integrating quantitative and qualitative data or findings, and missing opportunities to combine data in order to add rigour, enabling thorough and more complete analysis, provide credibility to results, and generate further important insights about the intervention under evaluation. This paper reports on a 2 day expert meeting funded by the United Kingdom Medical Research Council Hubs for Trials Methodology Research with the aims to identify current strengths and weaknesses in the integration of quantitative and qualitative methods in clinical trials, establish the next steps required to provide the trials community with guidance on the integration of mixed methods in RCTs and set-up a network of individuals, groups and organisations willing to collaborate on related methodological activity. We summarise integration techniques and go beyond previous publications by highlighting the potential value of integration using three examples that are specific to RCTs. We suggest that applying mixed methods integration techniques to data or findings from studies involving both RCTs and qualitative research can yield insights that might be useful for understanding variation in outcomes, the mechanism by which interventions have an impact, and identifying ways of tailoring therapy to patient preference and type. Given a general lack of examples and knowledge of these techniques, researchers and funders will need future guidance on how to undertake and appraise them.

## BACKGROUND

It is common to undertake qualitative research alongside randomised controlled trials (RCTs) when evaluating complex interventions.[1 2] Qualitative research can be used to explore the feasibility, acceptability and implementation of an intervention to help understand how it was effective or why it was not effective within the RCT, or to explore the conduct of the RCT to help improve recruitment or retention rates.[2] Qualitative research can be undertaken as part of a mixed methods process evaluation,[3] as a qualitative process evaluation, or as an embedded qualitative study alongside a fully powered or pilot RCT.[4] In studies like this, researchers have a number of datasets to analyse: outcome data from the RCT, quantitative process data, and qualitative process data. Researchers tend to analyse these datasets separately and then consider their findings separately within the discussion section of the final report to funders. Researchers rarely integrate quantitative and qualitative data or findings,[1] or use formal analytical techniques recommended within wider mixed methods research. This may be because researchers are not aware of existing integration techniques or do not see the value of these techniques to the context of RCTs. Unfortunately, they may be missing opportunities to fully use data that has taken years to collect at great cost that could generate further important insights about the intervention under evaluation.

## AIMS

In 2017, 20 researchers who generate evidence on the effectiveness of health interventions in healthcare and public health, and/or who have written methodological articles on integration in mixed methods research, came together in a 2 day meeting funded by the United Kingdom Medical Research Council Hubs for Trials Methodology Research. The aims of the meeting were to: i) identify current strengths and weaknesses in the integration of quantitative and qualitative methods in clinical trials; ii) establish the next steps required to provide the trials community with guidance on the integration

of mixed methods in RCTs; iii) set-up a network of individuals, groups and organisations willing to collaborate on related methodological activity.

The meeting was structured in the form of a summit (day 1) and expert panel (day 2) with pre-circulated agenda topics. The summit was focused on current strengths and weaknesses in the integration of quantitative and qualitative trial data including presentations from leading experts on integration and clinical trials, mixed methods analyses, mixed methods study designs, and writing mixed methods grant applications and study reports. Each topic was followed by facilitated small group discussions with focused questions for each group. These outputs were summarised and presented to the whole group on day two where the expert panel considered the need for guidance on the analytic integration of quantitative and qualitative trial data and the next steps required to produce it. Delegates also discussed the writing of a position paper, publication plans and future collaboration networks.

In this paper, the attendees at that meeting summarise relevant integration techniques, describe the key conclusions drawn, and propose the potential value of integration using three examples of integration suggested by the expert panel in the context of both pilot and fully powered RCTs. The focus is on RCTs of health interventions but the paper is likely to be relevant to RCTs in other fields such as education and social care.

### Summary of integration techniques

The term integration is used to describe an intentional process whereby researchers use quantitative and qualitative data or findings interdependently to address a common goal. Mixed methods researchers have described a range of integration techniques.[5–9] One key integration technique is joint displays that involve the production of a figure, table or graph to juxtapose and compare quantitative and qualitative data or findings. Other techniques can compare the qualitative responses of participants based on their measured response to the trial, or create a consolidated database drawing on both quantitative and (transformed) qualitative data for further statistical analysis.

### Meeting conclusions: the need for examples and guidance

Having listened to discussions about integration in mixed methods research, and considered its role in the context of RCTs, attendees concluded that integration rarely occurs in practice, that there is much to be learnt from integration techniques from wider mixed methods research, and that an important step is to communicate the potential value of integration to the research community through relevant examples. Once researchers see evidence of value, in terms of this practice generating credible new and useful insights, formal guidance could be developed to ensure the quality of this endeavour. Such guidance could include systematic reviews of the utility of different methods used for mixed methods data

integration with decision aids to help researchers make decisions about the most appropriate approach to adopt in different circumstances, how to ensure that the potential for data integration is factored into trial design at the earliest possible stages, the meaning of credibility for integrative practices, and how reviewers, funding boards, and readers can make judgments about the credibility of integration. Attendees agreed that in common with other clinical guidance documents, concrete examples of where these techniques have been used are required as part of the guideline. However, attendees struggled to identify many published examples of integration undertaken in the context of RCTs.

### Examples of the potential value of integration in the context of RCTs

Three examples were nonetheless identified and suggested by participants, and are described below. These examples integrate data or findings in the context of pilot or fully powered RCTs, and use various approaches to data integration and synthesis, guided by quantitative and/or qualitative data.

### Example 1: integrating findings from a pilot RCT and embedded qualitative interview study to develop insights about treatment responses

Thirty-one cancer patients took part in a cross-over pilot RCT of music medicine compared with music therapy.[10] Music medicine consisted of two sessions of listening to pre-recorded music; music therapy involved two sessions of interactive music-making with a music therapist. Before and after each session participants rated their mood, anxiety, relaxation and pain; 30 participants also completed a qualitative interview at the end of treatment. The quantitative pre-post change in outcomes and qualitative interview data of experiences were analysed separately. Findings were then integrated to explore why some individual patients appeared to benefit more from one intervention than the other. As a pilot RCT the study did not have sufficient statistical power to detect clinically meaningful differences in the effectiveness of these treatments, should such differences exist, but the study quantitative findings were that participants' mood, anxiety, relaxation and pain appeared to improve following either music medicine or music therapy. The findings from the qualitative interviews were that patients experienced music medicine and music therapy as relaxing and fun. Participants escaped from stress in general and worries related to cancer in particular. Music also offered hope for the future. In addition, music therapy enabled patients to be creative and the presence of a therapist helped some patients to release emotion. Other patients felt more comfortable with music medicine because there was no therapist and because they could listen to familiar music.

Findings were integrated during the analysis phase of the study to explore reasons for variation in outcomes. The researchers did this by creating a joint display of quantitative and qualitative findings for four groups of

**Table 4** Joint display of patient experiences per treatment benefits

| Treatment benefits | Change in music therapy[a] | Change in music medicine[a] | Patient experiences |
|---|---|---|---|
| ↑MT, ↓ MM | 0.65 to 1.88 | −0.11 to 0.38 | • Emphasize the importance of therapeutic relationship and support by therapist<br>• Enjoy the creative aspect of music making<br>• Are hopeful for the future |
| ↑MM, ↓ MT | −0.46 to 0.59 | 0.33 to 1.63 | • Apprehensive about active music making<br>• Prefer familiarity of pre-recorded music<br>• Hesitant about exploring feelings related to cancer |
| ↑MT, ↑ MM | 0.61 to 1.07 | 0.73 to 1.37 | • Strong conviction about the power of music to support and give hope<br>• Use music for mental escape<br>• Use music for emotional exploration and value processing of emotions with therapist |
| ↓ MT, ↓ MM | −0.67 to −1.03 | −0.52 to −1.06 | • Hold little hope for the future<br>• Music evokes sad and traumatic memories<br>• Feel inadequate regarding music making and singing<br>• Prefer aesthetics of original recordings |

↑ great improvement, ↓ less improvement or worsening

[a] Range of overall $z$-scores (average of $z$-scores for mood, anxiety, relaxation, and pain)

**Figure 1** Joint display table summarising findings from RCT and qualitative research (Bradt J, Potvin N, Kesslick a et al. supportive care in cancer 2015,[10] reproduced with permission).

the 30 patients with both qualitative and quantitative data: participants who showed quantitative (a) improvement following music therapy (MT) but much less or no improvement following music medicine (MM) (b) improvement following MM but much less or no improvement following MT, (c) improvement following both interventions, and (d) deterioration following both interventions. Improvement or deterioration was determined based on z-scores for changes in mood, anxiety, relaxation and pain for each patient. Patient experiences, expressed in the qualitative interviews, were summarised in the table for each of the four groups. These experiences differed by each of the four groups. Patients who described valuing the therapeutic relationship and creative elements of making music in the qualitative research benefited more from MT than MM. Patients who were apprehensive about active music making and exploring feelings related to cancer benefited more from MM than MT (see figure 1).

Integration of findings from independent quantitative and qualitative analyses identified that participant preferences and attitudes appeared to impact on treatment benefits. This generates the hypothesis that an effective intervention could be one where cancer patients are offered a choice of music therapy or music medicine based on their preferences. This hypothesis could then be tested in a fully powered RCT. This approach to analysis is not without challenges. Although blinding of qualitative researchers is often thought to be highly difficult because of the nature of the researcher/participant conversations, with careful organisation it is possible do initial analyses blind to knowledge of the other, for example by analysing qualitative findings not knowing the trial outcomes, generating key finding statements from each dataset still blind. Blinding of mixed methods analysts to the quantitatively determined group

allocations may be difficult but would be desirable during integration to further reduce the potential for analytical bias. Despite organisational difficulties, however, there is the potential for this technique to identify new insights about variation in outcomes and how to develop interventions to address them.

### Example 2: integrating data from a pilot RCT and qualitative process data to develop insights about treatment adherence and outcomes

Sixty-eight adults with depression participated in a pilot RCT of Morita Therapy plus Treatment As Usual (TAU) vs TAU alone.[11] Morita Therapy is a Japanese psychological therapy for common mental health problems where patients are taught that unpleasant thoughts and emotions ebb and flow as a matter of course, cannot be controlled by will, and can be accepted as part of the natural ecology of the human experience. Twenty-eight intervention participants also attended a post-treatment qualitative interview.[12] Following separate analysis of the RCT data and the qualitative interviews, data from the interviews and pilot RCT were integrated at the level of individual participants who had provided both types of data to explore how participants' views of the intervention related to the number of sessions they attended and whether they responded to treatment. Morita Therapy participants attended a mean of 8 treatment sessions out of a maximum of 12; 24/34 (70.6%) adhered to a per-protocol minimum dose (≥5 sessions). The pilot RCT was not powered to detect clinically meaningful differences in treatment effectiveness; however, at follow-up 22/33 intervention participants (66.7%) scored below the threshold for major depressive disorder on a depression symptom checklist compared with 13.3% of controls. During the

qualitative interviews, many but not all of those receiving Morita Therapy found it acceptable. Acceptability was related to participants' expectations and understandings of treatment (or 'orientation' towards treatment) being compatible or not with the principles of Morita Therapy. Participants distinguished between engaging with Morita Therapy on a conceptual level and engaging with it on a practical level such as finding the required time to do it.

Quantitative and qualitative data were then integrated for individual intervention participants. Qualitative data were organised into typologies of views on Morita Therapy, according to the extent to which participants found (1) the principles and (2) the practical processes of the therapy acceptable. Quantitative data from the RCT on the numbers of sessions attended and the clinical outcomes were added into a joint display. Participants who identified with Morita Therapy principles typically responded to treatment regardless of the number of sessions they attended; conversely, those whose orientation towards treatment was incompatible with the intervention did not respond to treatment, again regardless of treatment adherence. Participants whose personal circumstances (such as a lack of time or support from others) impeded their opportunity to engage in the intervention generally attended the fewest number of sessions but this did not drive clinical outcome.

By integrating qualitative and quantitative data the possibility of new relationships between orientation towards treatment, adherence and clinical outcomes were found. These suggested that personalising depression treatment by choosing an approach (for example, "Western" or "Eastern" treatment, directive or non-directive approaches) according to the extent to which the conceptual underpinnings of that treatment were compatible with a person's own worldview, could lead to better individual outcomes. It also raised the possibility that, where this congruence is present, patients with demanding personal circumstances which constrain their ability to engage in treatment may still benefit from a lower intensity version of the same treatment offered in fewer sessions. Clearly the small numbers in this study is a limitation and the analysis was not conclusive, but rather they raised interesting possibilities for testing in future RCTs of personalised medicine.

### Example 3: integrating findings from the quantitative and qualitative parts of a process evaluation to interpret the findings of a fully powered cluster RCT

A large, multi-country factorial cluster RCT was conducted to examine the effectiveness of interventions aimed at decreasing antibiotic prescribing for acute cough by general practitioners (GPs).[13] Interventions were: (i) GP communication skills training and use of a patient booklet; (ii) training in the provision and use of a C-re-active protein (CRP) test device; (iii) both interventions; or (iv) neither intervention. Each of the separate interventions led to decreases in antibiotic prescribing and the combination of both led to the largest decrease. The

mixed methods process evaluation, undertaken alongside the RCT, collected quantitative and qualitative data on patients' and GPs' views on prescribing antibiotics for acute cough and their experiences of the interventions. Quantitative self-report data were collected from 2886 patients and 346 GPs. Qualitative data were collected from interviews with 62 patients and 66 GPs. Data from the four different sources in the process evaluation were analysed separately. Findings from all four sources were then compared in an integrative analysis.[14] Based on the surveys, there was a high level of satisfaction with both interventions. GPs reported that the communication skills training, use of a patient booklet, and training with use of the CRP test helped them to reduce prescribing. Patients who received the booklet reported the highest levels of self-care enablement for their cough and awareness that taking antibiotics could be risky and harmful. Based on the qualitative interviews, GPs felt that the communication skills training gave them greater confidence in addressing patient expectations for an antibiotic and that the CRP test was helpful to decrease diagnostic uncertainty and reassure patients. The booklet and use of the CRP test were acceptable to patients and patients perceived that both the interventions supported GPs' prescribing decision.

The key findings from each of the four datasets in the process evaluation were integrated by summarising them in a form of joint display known as Triangulation Protocol.[15] Three analysts independently compared findings across the datasets, considering where they agreed, partly agreed, did not agree, or where there was an unexpected gap in findings from one of the datasets. Examples of the joint displays are given within the article. There was disagreement between findings from different datasets in terms of the utility of the CRP test. In the qualitative interviews with GPs, the CRP test was viewed as helping to convince patients that a no-antibiotic decision was appropriate when the patient expected an antibiotic to be prescribed. In contrast, patients reported in interviews that they were confident in the GP's prescription decision regardless of whether or not the CRP test had been undertaken, especially if they were given a detailed explanation by their GP and a booklet on self-care. Findings from the patient survey reinforced the importance of communication to patients. This highlighted the importance to patients of improved communication rather than diagnostics for antibiotic prescribing. Limitations included the non-complementary nature of the retrospectively designed interview guides and the lack of individual participant level data linking. This study also highlighted that attention needs to be paid to the quality of the application of integration techniques, for example the use of three independent analysts to create and interpret the joint display.

### CONCLUSIONS

Applying mixed methods integration techniques to data or findings from studies involving both RCTs and

qualitative research can yield insights that might be useful for understanding variation in outcomes, the mechanism by which interventions have an impact, and identifying ways of tailoring therapy to patient preference and type. The three examples given here are by no means definitive. However, they do illustrate different approaches to data integration including synthesis of findings driven by quantitative data (example 1), qualitative data (example 2), or by both equally (example 3). Further examples of integration using these and other analytical techniques might help to persuade researchers and funders of the value of doing this. However, published examples remain rare and the research community has no guidance on which of these and other techniques might yield the best added value to RCTs. The development of guidance will help to ensure the quality of this practice by, for example, recommending the use of independent data collection and analysis, blinding of analysts undertaking integration analyses, and rationale for the choice of analysis methods. Such guidance could include systematic reviews of the utility of different methods used for mixed methods data integration with decision aids to help. Guidance might also equip research funding boards, reviewers and other readers to judge the credibility of any integration.

#### Author affiliations
[1]Institute of Health Sciences, College of Medicine and Health, University of Exeter, Exeter, UK
[2]Transitional Research and Social Innovation Group, Western Sydney University, Penrith South, New South Wales, Australia
[3]Department of Care Science, Malmo University, Malmo, Skåne, Sweden
[4]Department of Nursing Education, Lovisenberg Diaconal University College, Oslo, Akershus, Norway
[5]MRC/CSO Social and Public Health Sciences Unit, University of Glasgow, Glasgow, UK
[6]Department of Biostatistics and Health Informatics, Institute of Psychiatry, Psychology and Neuroscience, King's College London, London, UK
[7]Population Health Sciences, University of Bristol, Bristol, UK
[8]Medical School, Swansea University, Swansea, UK
[9]School of Primary, Community and Social Care, Keele University, Keele, UK
[10]Nottingham Clinical Trials Unit, University of Nottingham, Nottingham, UK
[11]School of Social Sciences, Cardiff University, Cardiff, UK
[12]School of Education, University of Cincinnati College of Education Criminal Justice and Human Services, Cincinnati, Ohio, USA
[13]Department of Primary Care Health Sciences, and NIHR Health Protection Research Unit in Healthcare Associated Infections and Antimicrobial Resistance, University of Oxford, Oxford, UK
[14]Institute of Health and Wellbeing, University of Glasgow, Glasgow, UK
[15]Institute of Population Health Sciences, University of Liverpool, Liverpool, UK
[16]Medical Care Research Unit, School of Health and Related Research, University of Sheffield, Sheffield, UK

**Acknowledgements** Holly Sugg provided information for example 2. DR is supported by the National Institute for Health Research (NIHR) Applied Research Collaboration (ARC) South West Peninsula. CJ is supported by the National Institute for Health Research (NIHR) Applied Research Collaboration (ARC) West Midlands. JH is supported by the National Institute for Health Research (NIHR) Applied Research Collaboration (ARC) West. STC is funded by the National Institute for Health Research Health Protection Research Unit (NIHR HPRU) in Healthcare Associated Infections and Antimicrobial Resistance at the University of Oxford in partnership with Public Health England (PHE) [HPRU-2012-10041]. The views expressed are those of the author(s) and not necessarily those of the MRC, NHS, the NIHR, the Department of Health and Social Care or Public Health England.

**Contributors** DAR, JH and AOC conceived, planned and conducted the meeting. AOC, JH and DAR drafted the original manuscript. PB, GB, PC, RE, JF, JH, HAH, CJ, AM, GM, VLPC, ST-C, JW, FCW, SW, BY attended the meeting and read, edited and approved the final manuscript.

**Funding** The United Kingdom Medical Research Council Hubs for Trials Methodology Research (HTMR) Network funded the meeting. The funder had no role in the design of the meeting and in writing the manuscript.

**Competing interests** None declared.

**Patient consent for publication** Not required.

**Provenance and peer review** Not commissioned; externally peer reviewed.

**ORCID iDs**
David A Richards http://orcid.org/0000-0002-8821-5027
Julia Frost http://orcid.org/0000-0002-3503-5911
Hayley Anne Hutchings http://orcid.org/0000-0003-4155-1741
Vicki L Plano Clark http://orcid.org/0000-0002-9709-7982
Alicia O'Cathain http://orcid.org/0000-0003-4033-506X

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
