## [Reviewer comments · BMJ Open]

ARTICLE DETAILS

TITLE (PROVISIONAL)	Integrating quantitative and qualitative data and findings when undertaking randomised controlled trials
AUTHORS	Richards, David; Bazeley, Patricia; Borglin, Gunilla; Craig, Peter; Emsley, Richard; Frost, Julia; Hill, Jacqueline; Horwood, Jeremy; Hutchings, Hayley; Jinks, Clare; Montgomery, Alan; Moore, Graham; Plano Clark, Vicki; Tonkin-Crine, Sarah; Wade, Julia; Warren, Fiona; Wyke, Sally; Young, Bridget; O'Cathain, Alicia

VERSION 1 - REVIEW

REVIEWER	Pierre Pluye McGill University
REVIEW RETURNED	09-Aug-2019

GENERAL COMMENTS	Thanks for inviting me to review this manuscript. Overall, it concerns an increasingly important methodological issue in health sciences. The international team of authors is impressive, and combines all necessary expertise in numerous methods, including RCTs, qualitative research and mixed methods. The manuscript reports a 2-day workshop with three exemplar studies. My main suggestion is that authors might consider developing the introduction and the conclusion (add 2 or 3 sentences each) to clarify/highlight the justification of this paper and its contribution to knowledge, respectively (compared to existing reviews and guidance, e.g., the O'Cathain et al.'s practical guide). Other comments are listed below. P. 3. Abstract: Reformulate "informally" as this seems contradictory with a formal discussion section in a published article (see details below). P. 4. Line 24: Same comment. I understand what authors mean, but this sentence is crucial in justifying the manuscript, while it seems somewhat unclear or paradoxical. Indeed literally speaking, something cannot be "informal" when it is published in an article (as mentioned above). In addition, "informal integration" (whatever it is) constitutes a form of integration at least from a reader's point of view (in plain fact, the sentence includes the word "integrate"), while the prescriptive tone of the manuscript suggests authors claim that methods "informally integrated" are poorly or not integrated. This strong authoritative claim needs more explanations and references, which will help readers with no or little training or experience in mixed methods research. My 5 cents: I suggest authors replace "informal integration" with something else when they feel that this does not meet criteria of
--

integration. In contrast to the three proposed exemplar studies, an ideal-type example (no name) of such “informal integration / no proper integration” stance would help.

P. 5. Line 39: Add the aim/purpose of the manuscript. This has been displayed in the abstract, but seems to be missing in the body of the text.

P. 5. Lines 44-53: Authors suggest this work may ultimately lead to method guidance. The term “guidance” is reused in conclusion. Thus, authors might integrate this in the aim/purpose of this manuscript. Authors report a workshop and 3 exemplar studies, while they may have (rightly) more ambition in terms of paving the way toward the development of method guidance for health, education and social sciences. Adding 1 or 2 sentences on future steps, e.g., a systematic methodological review, can strengthen the conclusion.

P. 6. Line 8-12: I agree. We (a postdoctoral fellow and myself) found only a couple of examples for our mixed methods courses, while there are multiple examples of qualitative research performed alongside RCTs without integration of QUAL and QUAN components at any level (no integration in terms of worldviews, literature review, research questions, design, data collection/analysis, and results).

PP. 6-7. Example 1: Great choice. I like the Bradt's study, and provide it to my trainees in mixed methods courses as an exemplar integration and clear reporting of mixed methods. I agree with the Authors' summary. However, I am not convinced that “blinding of qualitative researchers to the groups” would be pertinent and feasible in this study. Remove, or replace this limitation with another, or justify and develop the claim that such blinding “would be important”. As mentioned in the manuscript, the qualitative interviews were conducted after the intervention time-period; thus, they did not affect the outcome measurement. In addition, they scrutinized patients' experience, which included the type of treatment. Patients cannot conceal their treatment during interviews even though they were asked to hide the treatment name from interviewers.

Stated otherwise, the limitations of mixing QUAL and QUAN components in RCTs are essential, but I am not convinced that this can be addressed using a single example in such short commentary manuscript. To be fair, authors should add at least a limitation for each presented study. Another option may be to remove this specific limitation (controversial), and simply list all types of limitations of mixing QUAL and QUAN components in RCTs in conclusion.

P. 8. Example 2: Spell the meaning of “TAU” at first occurrence.

P. 8. Lines 21 and 53: Explain the integration at the “level of individual participants” because the number of participants in the intervention group (n=33) does not match the number of interviewees (n=28).

P.9. Line 39: Replace “prove that these insights were true” with something more nuanced (toned down), e.g., “support these qualitative insights in a statistical manner”.

	Reference #11 and #12 seem similar: Replace reference #12 with Sugg et al's paper in BMJ open 2019 entitled "an embedded qualitative etc." P. 11. Conclusion (lines 30-33): Consider adding "data collection and" between "independent" and "analysts" and replacing "blinding of analysts" with "blinding of statisticians". I am not convinced that qualitative researchers usually can be blinded (similar somewhat to surgeons conducting RCTs).
--	--

REVIEWER	David Henry Institute for Evidence Based Healthcare Bond University Gold Coast, Queensland, Australia
REVIEW RETURNED	15-Sep-2019

GENERAL COMMENTS	I have asked this a 'major revision'. I accept its not possible to revise the details and conclusions of a meeting but I think the authors should address the concerns expressed in my review. This is a communication article describing a meeting to discuss integration of qualitative and quantitative data collected during randomised trials. Before discussing the article in more detail I will address the journal criteria for consideration of an article of this type  • Are the issues raised by the article important to BMJ Open's broad and international readership that includes patients, researchers, policy makers, health professionals, and doctors of all disciplines?  o Yes the general topic is important and is of interest to a broad range of stakeholders • Is the article interesting and offering novel insights that have not been sufficiently considered in the existing published literature? Is the article well written and is the content clearly presented? Does it have a clear message?  o The article is of some interest and is clearly written. I have some criticisms of features (or lack of) that reduce the insights that can be drawn from the paper. These are discussed below • Will the article help medical researchers, patients or related groups of readers to make better decisions?  o It may prompt researchers to consider collection of qualitative data during planning of randomised trials but the report is limited in its advice about how these data might be analysed and used • Does the article demonstrate one or more of the following values: transparency, openness, collaboration, innovation, reproducibility, patient/ public involvement, improving peer review and journal best practice, and reducing research waste  o This criterion is not really applicable to this article I think the topic is important and neglected. It is certainly worth addressing in some detail and to this end the authors held a 2-day
---

	meeting. In the article they provide some background to the topic and describe integration as “joint displays that involve the production of a figure, table or graph to bring together and compare quantitative and qualitative data or findings.” This is an interesting area of study. The resulting report is concise but its quite superficial, lacks much detail and does not provide much additional insights. My main criticisms are: 1) It is not clear if a systematic or scoping review of the topic was conducted in advance of the meeting. I am assuming not 2) It is not clear if any technical documents or examples of data integration were circulated to participants ahead of the meeting. 3) The meeting itself is not described in any detail. I can’t tell if any structured processes were used to identify key issues, deepen the discussion or achieve any consensus 4) I am not clear how the meeting participants came up with 3 examples used to illustrate integration techniques 1) The examples are described but there is no analysis or comparison of the techniques that were used. I understand that this is limited by the small number of articles but it is possible to contrast the approaches. For instance, in example 1 a crossover trial of music medicine and music therapy it was possible to categorise participants as falling into 4 groups and compare the qualitative findings. In example 2 participants were categorized by their qualitative response – engaging at a conceptual level or engaging at a practical level – and this was used to compare treatment responses. In example 3 (Interventions to reduce antibiotic prescribing) the quantitative data were not analysed by response but by treatment allocation group and between patients and GPs. So these 3 examples illustrate a number of different approaches to integrating qualitative and quantitative data and discussion of these could have provided some insights and guidance to future work. I realise that the authors cant rewrite the meeting record but circulation of a post-meeting analysis could have elicited additional responses 5) Finally the Conclusions seem brief and superficial and don’t provide guidance on the directions of future work. As with the previous point the authors cant editorialise the report, but circulation of a post meeting summary of conclusions could have deepened this section considerably.
--	---

VERSION 1 – AUTHOR RESPONSE

Reviewer: 1

Reviewer Name: Pierre Pluye

Institution and Country: McGill University

Please state any competing interests or state ‘None declared’: none

Please leave your comments for the authors below

Thanks for inviting me to review this manuscript. Overall, it concerns an increasingly important methodological issue in health sciences. The international team of authors is impressive, and combines all necessary expertise in numerous methods, including RCTs, qualitative research and

mixed methods. The manuscript reports a 2-day workshop with three exemplar studies. My main suggestion is that authors might consider developing the introduction and the conclusion (add 2 or 3 sentences each) to clarify/highlight the justification of this paper and its contribution to knowledge, respectively (compared to existing reviews and guidance, e.g., the O’Cathain et al.’s practical guide). Other comments are listed below.

THANK YOU FOR THIS COMMENT. WE HAVE ADDED FURTHER TEXT THROUGHOUT THE PAPER TO PROVIDE MORE JUSTIFICATION OF OUR MEETING AND SPECIFIED THAT OUR FOCUS HERE WAS ON RCT DESIGNS SPECIFICALLY.

P. 3. Abstract: Reformulate “informally” as this seems contradictory with a formal discussion section in a published article (see details below).

SENTENCE AMENDED AS REQUESTED

P. 4. Line 24: Same comment. I understand what authors mean, but this sentence is crucial in justifying the manuscript, while it seems somewhat unclear or paradoxical. Indeed literally speaking, something cannot be “informal” when it is published in an article (as mentioned above). In addition, “informal integration” (whatever it is) constitutes a form of integration at least from a reader’s point of view (in plain fact, the sentence includes the word “integrate”), while the prescriptive tone of the manuscript suggests authors claim that methods “informally integrated” are poorly or not integrated. This strong authoritative claim needs more explanations and references, which will help readers with no or little training or experience in mixed methods research. My 5 cents: I suggest authors replace “informal integration” with something else when they feel that this does not meet criteria of integration. In contrast to the three proposed exemplar studies, an ideal-type example (no name) of such “informal integration / no proper integration” stance would help.

WE HAVE USED THE SAME FORM OF WORDS HERE AS IN THE ABSTRACT TO REMOVE THE LITERAL CONFUSION

P. 5. Line 39: Add the aim/purpose of the manuscript. This has been displayed in the abstract, but seems to be missing in the body of the text.

THE AIM WAS/IS INCLUDED IN THE FINAL PARAGRAPH ON PAGE 4. WE HAVE REVISED THE TITLE OF THIS SECTION TO MAKE THIS CLEAR AND EXPANDED THE DESCRIPTION OF THE AIMS IN BOTH THE ABSTRACT AND BODY OF THE PAPER

P. 5. Lines 44-53: Authors suggest this work may ultimately lead to method guidance. The term “guidance” is reused in conclusion. Thus, authors might integrate this in the aim/purpose of this manuscript. Authors report a workshop and 3 exemplar studies, while they may have (rightly) more

ambition in terms of paving the way toward the development of method guidance for health, education and social sciences. Adding 1 or 2 sentences on future steps, e.g., a systematic methodological review, can strengthen the conclusion.

WE HAVE REFERRED TO THE 'NEED FOR GUIDANCE' IN THE AIM OF THE MEETING. WE HAVE REFERRED TO REVIEWS AND DECISION AIDS AS A COMPONENT OF SUCH GUIDANCE

P. 6. Line 8-12: I agree. We (a postdoctoral fellow and myself) found only a couple of examples for our mixed methods courses, while there are multiple examples of qualitative research performed alongside RCTs without integration of QUAL and QUAN components at any level (no integration in terms of worldviews, literature review, research questions, design, data collection/analysis, and results).

THANK YOU FOR THIS COMMENT

PP. 6-7. Example 1: Great choice. I like the Bradt's study, and provide it to my trainees in mixed methods courses as an exemplar integration and clear reporting of mixed methods. I agree with the Authors' summary. However, I am not convinced that "blinding of qualitative researchers to the groups" would be pertinent and feasible in this study. Remove, or replace this limitation with another, or justify and develop the claim that such blinding "would be important". As mentioned in the manuscript, the qualitative interviews were conducted after the intervention time-period; thus, they did not affect the outcome measurement. In addition, they scrutinized patients' experience, which included the type of treatment. Patients cannot conceal their treatment during interviews even though they were asked to hide the treatment name from interviewers.

THANK YOU FOR THIS COMMENT AND FOR CONFIRMING OUR DECISION MAKING. HOWEVER, WE DISAGREE ABOUT BLINDING OF RESEARCHERS. WE AGREE THAT IT IS VERY DIFFICULT TO BLIND QUALITATIVE RESEARCHERS TO GROUP ALLOCATION BUT IT IS POSSIBLE TO BLIND THEM TO OVERALL TRIAL OUTCOMES. WE WERE SUGGESTING THAT THOSE UNDERTAKING THE MIXED METHODS ANALYSES COULD DO SO BLINDED TO THE GROUP DEFINITIONS (EITHER QUALITATIVELY OR QUANTITATIVELY DERIVED) AND ALLOCATIONS SINCE IN THIS CASE THESE WERE A MIX OF TREATMENT ALLOCATION AND OUTCOMES. CLEARLY, DURING THE ANALYSES ANALYSTS MAY 'GUESS' TREATMENT ALLOCATION AND OUTCOME EXPERIENCES AND EVEN THE ANALYTICAL GROUP ALLOCATIONS, BUT IT WOULD BE GOOD PRACTICE TO START THE PROCESS BLINDED, SIMILAR TO THE MANNER IN WHICH STATISTICAL ANALYSTS UNDERTAKE THEIR WORK ON DATA ALREADY COLLECTED (WHERE UNBLINDING MAY HAVE OCCURRED UNWITTINGLY) PRIOR TO ANALYST UNBLINDING. WE HAVE REWRITTEN THE PARAGRAPH TO EXPLAIN OUR MEANING MORE THOROUGHLY.

Stated otherwise, the limitations of mixing QUAL and QUAN components in RCTs are essential, but I am not convinced that this can be addressed using a single example in such short commentary manuscript. To be fair, authors should add at least a limitation for each presented study. Another

option may be to remove this specific limitation (controversial), and simply list all types of limitations of mixing QUAL and QUAN components in RCTs in conclusion.

WE ACCEPT THAT THIS IS A DILEMMA. WE HAVE PREFERRED TO RETAIN OUR APPROACH AS EXEMPLARS RATHER THAN FULL LISTINGS OF METHODOLOGICAL PROBLEMS, SINCE THESE WOULD BE EXPLORED FULLY IN FUTURE GUIDANCE. AS SUGGESTED BY THE REVIEWER, WE HAVE ENSURED ALL EXAMPLES INCLUDE SOME REFERENCE TO LIMITATIONS

P. 8. Example 2: Spell the meaning of "TAU" at first occurrence.

THIS WAS ALREADY INCLUDED IN THE MANUSCRIPT. WE HAVE CAPITALISED IT TO MAKE IT MORE OBVIOUS

P. 8. Lines 21 and 53: Explain the integration at the "level of individual participants" because the number of participants in the intervention group (n=33) does not match the number of interviewees (n=28).

WE HAVE REWRITTEN THIS TO MAKE IT CLEAR THAT DATA CAN ONLY BE INTEGRATED AT THE LEVEL OF THE INDIVIDUAL IF BOTH TYPES OF DATA HAVE BEEN COLLECTED FROM INDIVIDUAL PARTICIPANTS

P.9. Line 39: Replace "prove that these insights were true" with something more nuanced (toned down), e.g., "support these qualitative insights in a statistical manner".

WE HAVE REWRITTEN THIS SENTENCE

Reference #11 and #12 seem similar: Replace reference #12 with Sugg et al's paper in BMJ open 2019 entitled "an embedded qualitative etc."

THANK YOU FOR SPOTTING THIS ERROR. WE HAVE REPLACED THE REFERENCE AS SUGGESTED

P. 11. Conclusion (lines 30-33): Consider adding "data collection and" between "independent" and "analysts" and replacing "blinding of analysts" with "blinding of statisticians". I am not convinced that qualitative researchers usually can be blinded (similar somewhat to surgeons conducting RCTs).

AS DISCUSSED UNDER THE PREVIOUS COMMENT, WE SUGGEST THAT IT MAY BE BOTH POSSIBLE IN SOME CIRCUMSTANCES AND DESIRABLE TO BLIND GROUP ALLOCATIONS TO INTEGRATIVE ANALYSTS. WE HAVE MADE THIS CLEAR IN THE CONCLUSION IN A SIMILAR MANNER TO OUR PREVIOUS RESPONSE IN REFERENCE TO EXAMPLE 1.

Reviewer: 2

Reviewer Name: David Henry

Institution and Country: Institute for Evidence Based Healthcare Bond University Gold Coast, Queensland, Australia

Please state any competing interests or state 'None declared': None

Please leave your comments for the authors below

I have marked this a 'major revision'. I accept its not possible to revise the details and conclusions of a meeting but I think the authors should address the concerns expressed in my review.

THANK YOU FOR THIS COMMENT AND FOR UNDERSTANDING THE CONSTRAINTS OF OUR ARTICLE. WE HOPE THE RESPONSES BELOW SATISFY THE REVISION SUGGESTIONS

This is a communication article describing a meeting to discuss integration of qualitative and quantitative data collected during randomised trials. Before discussing the article in more detail I will address the journal criteria for consideration of an article of this type

- Are the issues raised by the article important to BMJ Open's broad and international readership that includes patients, researchers, policy makers, health professionals, and doctors of all disciplines?
 - o Yes the general topic is important and is of interest to a broad range of stakeholders

THANK YOU, WE AGREE

- Is the article interesting and offering novel insights that have not been sufficiently considered in the existing published literature? Is the article well written and is the content clearly presented? Does it have a clear message?
 - o The article is of some interest and is clearly written. I have some criticisms of features (or lack of) that reduce the insights that can be drawn from the paper. These are discussed below

WE DESCRIBE OUR RESPONSES IN DETAIL BELOW

- Will the article help medical researchers, patients or related groups of readers to make better decisions?

- o It may prompt researchers to consider collection of qualitative data during planning of randomised trials but the report is limited in its advice about how these data might be analysed and used

WE HOPE WE EXPLAIN THIS THOROUGHLY IN OUR REVISIONS AS THIS IS NOT INTENDED AS AN ADVISORY DOCUMENT AT THIS STAGE IN THE DEVELOPMENT OF THESE METHODS

- Does the article demonstrate one or more of the following values: transparency, openness, collaboration, innovation, reproducibility, patient/ public involvement, improving peer review and journal best practice, and reducing research waste

- o This criterion is not really applicable to this article

I think the topic is important and neglected. It is certainly worth addressing in some detail and to this end the authors held a 2-day meeting. In the article they provide some background to the topic and describe integration as “joint displays that involve the production of a figure, table or graph to bring together and compare quantitative and qualitative data or findings.” This is an interesting area of study. The resulting report is concise but its quite superficial, lacks much detail and does not provide much additional insights. My main criticisms are:

1) It is not clear if a systematic or scoping review of the topic was conducted in advance of the meeting. I am assuming not

WE APPRECIATE THE VIEW THAT THIS TOPIC IS IMPORTANT; WE OF COURSE AGREE.

THE REVIEWER'S ASSUMPTIONS ARE CORRECT. THIS WAS A MEETING OF EXPERTS FROM THE UK AND INTERNATIONALLY WHO WERE ALL EXPERIENCED IN THE AREA, MANY OF WHOM HAD CONTRIBUTED SIGNIFICANTLY TO THE LITERATURE. OUR AIM WAS TO MEET FOR THE FIRST TIME AS A GROUP AND DEBATE THE FIELD AND DECIDE IF ISSUING GUIDANCE SIMILAR TO THOSE PUBLISHED ON OTHER AREAS OF COMPLEX INTERVENTIONS METHODS WOULD BE A REASONABLE RESPONSE TO THE LEVEL OF METHODOLOGICAL UNCERTAINTY IN THE AREA. SUBSEQUENT WORK MIGHT, OF COURSE, INCLUDE A REVIEW AS PART OF GUIDANCE. WE HAVE CLARIFIED THAT SUCH A REVIEW MIGHT BE UNDERTAKEN ON PAGE 6 OF THIS REVISION AND IN OUR STRENGTHENED CONCLUSIONS SECTION.

2) It is not clear if any technical documents or examples of data integration were circulated to participants ahead of the meeting.

THE MEETING WAS HIGHLY STRUCTURED AS WE WERE WANTING TO MAXIMISE THE CONTRIBUTION OF ALL CONCERNED. TIMETABLES AND DISCUSSION BRIEFS WERE CIRCULATED. THE DEFINITION OF DATA INTEGRATION DESCRIBED IN THIS PAPER WAS REFINED DURING THE MEETING'S DISCUSSIONS.

3) The meeting itself is not described in any detail. I can't tell if any structured processes were used to identify key issues, deepen the discussion or achieve any consensus

SEE ABOVE. WE HAVE ADDED A FULL DESCRIPTION OF THE MEETING PROCESS TO THE AIMS SECTION

4) I am not clear how the meeting participants came up with 3 examples used to illustrate integration techniques

1) The examples are described but there is no analysis or comparison of the techniques that were used. I understand that this is limited by the small number of articles but it is possible to contrast the approaches. For instance, in example 1 a crossover trial of music medicine and music therapy it was possible to categorise participants as falling into 4 groups and compare the qualitative findings. In example 2 participants were categorized by their qualitative response – engaging at a conceptual level or engaging at a practical level – and this was used to compare treatment responses. In example 3 (Interventions to reduce antibiotic prescribing) the quantitative data were not analysed by response but by treatment allocation group and between patients and GPs. So these 3 examples illustrate a number of different approaches to integrating qualitative and quantitative data and discussion of these could have provided some insights and guidance to future work. I realise that the authors cant rewrite the meeting record but circulation of a post-meeting analysis could have elicited additional responses

THESE SUGGESTIONS WERE MADE BY MEETING PARTICIPANTS (WITH DIFFICULTY GIVEN THAT DELEGATES ACKNOWLEDGED THE SPARSITY OF INTEGRATIVE EXAMPLES) IN ORDER TO PROVIDE THE READERS OF THIS ARTICLE WITH SOMETHING WITH WHICH TO HELP THEM UNDERSTAND THE DIFFERENCE BETWEEN MERELY DISCUSSING MIXED METHODS FINDINGS AND ACTUALLY INTEGRATING THEM AT THE LEVEL OF THE INDIVIDUAL STUDY PARTICIPANT. THEY ARE NOT CHOSEN TO ILLUSTRATE DEFINITE ANALYTICAL POSITIONS BUT WE AGREE WITH THE REVIEWER THAT AS THEY DO OUTLINE SPECIFIC APPROACHES, SOME FURTHER COMPARATIVE TEXT IS HELPFUL. WE HAVE ADDED MORE TEXT TO OUR CONCLUSION TO THIS POINT, AND MADE IT CLEAR THAT THESE EXAMPLES WERE GENERATED DURING THE COURSE OF OUR DISCUSSIONS

5) Finally the Conclusions seem brief and superficial and don't provide guidance on the directions of future work. As with the previous point the authors cant editorialise the report, but circulation of a post meeting summary of conclusions could have deepened this section considerably.

WE DID INDEED WRITE A REPORT AS A CONDITION OF OUR FUNDING. OUR CONCLUSIONS WERE NOT 'DEEPER' SINCE OUR MAIN DECISION WAS THAT CONSIDERABLE WORK WAS REQUIRED TO ADDRESS THE UNCERTAINTY AND LACK OF KNOWLEDGE IN THE AREA FOR THE RESEARCH COMMUNITY. NONETHELESS, WE HAVE TRIED TO STRENGTHEN THE PAPER'S CONCLUSION WITHOUT STEPPING OUTSIDE THE BOUNDARIES OF OUR MEETING OUTPUTS

VERSION 2 – REVIEW

REVIEWER	David Henry Bond University, Gold Coast, Queensland, Australia
REVIEW RETURNED	06-Nov-2019

GENERAL COMMENTS	I think my concerns have been well addressed in the revised manuscript - at least as far is possible, given the constraints of a meeting report. So I have no further suggestions for change
--